# Differential Gene Expression of *Mucor lusitanicus* under Aerobic and Anaerobic Conditions

**DOI:** 10.3390/jof8040404

**Published:** 2022-04-15

**Authors:** Mónika Homa, Sandugash Ibragimova, Csilla Szebenyi, Gábor Nagy, Nóra Zsindely, László Bodai, Csaba Vágvölgyi, Gábor Nagy, Tamás Papp

**Affiliations:** 1Department of Microbiology, Faculty of Science and Informatics, University of Szeged, H-6726 Szeged, Hungary; homamoni@gmail.com (M.H.); ibragimova_sandu@mail.ru (S.I.); szebecsilla@gmail.com (C.S.); csaba@bio.u-szeged.hu (C.V.); nagygab86@gmail.com (G.N.); 2Department of Biochemistry and Molecular Biology, Faculty of Science and Informatics, University of Szeged, H-6726 Szeged, Hungary; gbrngy@gmail.com (G.N.); bodai@bio.u-szeged.hu (L.B.); 3Department of Genetics, Faculty of Science and Informatics, University of Szeged, H-6726 Szeged, Hungary; zsindizsn@yahoo.com

**Keywords:** Mucorales, anaerobiosis, yeast-like growth, dimorphism, transporter, CAZymes

## Abstract

*Mucor lusitanicus* and some other members of the fungal order Mucorales display the phenomenon of morphological dimorphism. This means that these fungi aerobically produce filamentous hyphae, developing a coenocytic mycelium, but they grow in a multipolar yeast-like form under anaerobiosis. Revealing the molecular mechanism of the reversible yeast-hyphal transition can be interesting for both the biotechnological application and in the understanding of the pathomechanism of mucormycosis. In the present study, transcriptomic analyses were carried out after cultivating the fungus either aerobically or anaerobically revealing significant changes in gene expression under the two conditions. In total, 539 differentially expressed genes (FDR < 0.05, |log_2_FC| ≥ 3) were identified, including 190 upregulated and 349 downregulated transcripts. Within the metabolism-related genes, carbohydrate metabolism was proven to be especially affected. Anaerobiosis also affected the transcription of transporters: among the 14 up- and 42 downregulated transporters, several putative sugar transporters were detected. Moreover, a considerable number of transcripts related to amino acid transport and metabolism, lipid transport and metabolism, and energy production and conversion were proven to be downregulated when the culture had been transferred into an anaerobic atmosphere.

## 1. Introduction

*Mucor lusitanicus* (also known as *M. circinelloides* f. *lusitanicus*) is a filamentous fungus belonging to the order Mucorales [1]. Several *Mucor* species have biotechnological significance as producers of extracellular enzymes [2], carotenoids [3], unsaturated fatty acids, and organic acids [4,5,6]. Certain species, such as members of the genera *Rhizopus* and *Mucor*, are also known as agents of frequently fatal opportunistic fungal infections, the so-called mucormycoses [7,8]. Such infections associated with rapid progression and high mortality rates most frequently occur in immunocompromised patients and those with diabetic ketoacidosis and are notorious for the difficulties in their diagnosis and treatment [7,9]. Furthermore, *M. lusitanicus* and the closely related *M. circinelloides* (i.e., *M. circinelloides* f. *circinelloides*) are frequently used as a model organism during studies among others on carotenoid biosynthesis [10,11], biofuel production [12,13], morphogenesis [14,15,16], RNA interference [17], and pathogenicity mechanisms [18,19,20], as well as the molecular mechanisms of all these biological processes [21,22,23].

*Mucor lusitanicus*, similarly to several other mucoralean fungi, has the interesting ability of morphological dimorphism [14,24]. This fungus aerobically produces filamentous hyphae developing a coenocytic mycelium, but it grows in a multipolar yeast-like form under anaerobiosis [24,25]. Blockage of the mitochondrial energy-producing functions, such as electron transfer or oxidative phosphorylation, can also induce yeast-like growth, even under aerobic conditions [26]. The morphological transitions reversibly follow the changes of the environmental conditions [27].

cAMP-dependent protein kinase A (PKA) and calcineurin have been found as the main factors in the regulation of the dimorphic transition [15,16,26,28,29]. The addition of cAMP to the cultures can induce yeast-like growth even under aerobic conditions and the amount of PKA increases in the yeast-like cells, indicating that PKA may have a special role in the induction and/or maintenance of the yeast morphology [15]. At the same time, calcineurin plays a crucial role in the yeast-hyphal transition, and in the case of its deletion or inhibition, yeast cells fail to switch to hyphal growth [16].

Control and optimization of fungal morphology and growth are essential for various fermentation-based applications [24,25]. Morphological dimorphism may also have consequences on virulence, and it is regarded as a virulence factor for several pathogenic fungi [30]. In the case of *M. lusitanicus*, Lee et al. [16] found that the yeast-like form is less virulent than the spores or the hyphal form and raised the possibility of targeting the morphological transition to address the infection (e.g., by using calcineurin inhibitors) [16,20]. Thus, revealing the molecular mechanism of the yeast-hyphal transition can be interesting for both the potential biotechnological applications and developments aimed at controlling mucormycosis. In the present study, transcriptomic analyses were carried out after cultivating the fungus either aerobically or anaerobically. Transcriptomes were compared and genes differentially expressed were analyzed.

## 2. Materials and Methods

### 2.1. Strain and Culture Conditions

The *M. lusitanicus* double auxotrophic strain MS12 (*leuA^−^* and *pyrG^−^*), a derivative of the strain CBS 277.49 [31], was used in the present study. For spore collection, the strain was grown for 7 days on malt extract agar (MEA; 1% glucose, 0.25% yeast extract, 5% malt extract, 2% agar; pH 5.4) plates at 25 °C; then, the spores were washed with PBS by gentle scraping, and the spore suspensions obtained were maintained at 4 °C until use. For aerobic growth, 30 mL liquid minimal medium (YNB; 1% glucose, 0.15% ammonium sulphate, 0.15% sodium glutamate, 0.05% yeast nitrogen base without amino acids (BD Difco, Becton Dickinson, Franklin Lakes, NJ, USA), supplemented with 0.05% leucine and 0.05% uracil; pH 6.8) was inoculated to obtain a final concentration of 10^4^ sporangiospores/mL. The cultures were kept at 25 °C for 24 h with constant shaking (i.e., 190 rpm). For the mycelium–yeast transition, the 24 h old culture grown aerobically was transferred into a 50 mL flask into 50 mL fresh YNB medium and placed for 4 h into a BBL GasPak Anaerobic System (Becton Dickinson, Franklin Lakes, NJ, USA) at 25 °C.

### 2.2. RNA Extraction and Sequencing

For RNA purification, the 30 mL cultures detailed above were filtered through 0.45 µm MCE membrane filters (Millipore, Burlington, MA, USA) and the extraction was performed by using the Direct-zol™ RNA MiniPrep Kit (Zymo Research, Irvine, CA, USA), following the recommendations of the manufacturer. The samples were kept at −80 °C until their use. The quality of total RNA samples was analyzed by capillary gel electrophoresis in an Agilent 2100 Bioanalyzer using Agilent RNA 6000 Nano Kit and only samples with RIN > 9 were processed further. RNA-seq libraries were prepared from three biological and two technical replicates per treatment condition (i.e., six samples per treatment were used in total). Thus, six libraries were generated and all of them were sequenced. Libraries were pooled before sequencing, but since these were indexed, libraries sequence reads belonging to different samples could be demultiplexed (separated to different files) after sequencing. From 800 ng total RNA samples, polyA-RNA was isolated using NEBNext Poly(A) mRNA Magnetic Isolation Module (New England Biolabs, Ipswich, MA, USA), then indexed; strand-specific sequencing libraries were prepared using NEBNext Ultra II Directional RNA Library Prep Kit (New England Biolabs, Ipswich, MA, USA) following the protocol of the manufacturer. Sequencing libraries were validated and quantitated with an Agilent 2100 Bioanalyzer using Agilent DNA 1000 Kit; the average fragment length of the libraries was in the range of 271–289 bp. Libraries were pooled in equimolar concentrations, and after denaturing, the library pool was loaded at 15 pM concentration with 1% PhiX Control V3 (Illumina, San Diego, CA, USA) in a MiSeq Reagent Kit v3 (150-cycle) kit and sequenced in an Illumina MiSeq instrument producing 2 × 75 bp paired-end reads.

### 2.3. Analysis of the RNA-Seq Data

FASTQ files were generated with the GenerateFASTQ 1.1.0.64 application on Illumina BaseSpace. Sequence quality checks and adapter trimming were performed using TrimGalore/Cutadapt. Sequence reads were aligned to the *M. lusitanicus* CBS 277.49 v2.0 reference genome assembly available from the Joint Genome Institute (JGI) website (https://genome.jgi.doe.gov/Mucci2/Mucci2.home.html (accessed on 5 March 2021) [32]) using HISAT2 (parameters: –max-intronlen 45000, –rna-strandness RF). From the generated binary alignment files, gene specific read counts were calculated in R using the summarizeOverlaps (parameters: ignore.strand = FALSE, singleEnd = FALSE, fragments = TRUE, preprocess.reads = invertStrand) Bioconductor package with a TxDb transcript metadata object generated from the transcript annotation available from JGI using the makeTxDbFromGFF function of the GenomicFeatures Bioconductor package. Differential gene expression analysis was performed with DESeq2 parameter: *p* < 0.05, and a cpm > 1 read count filter. Gene Ontology analysis was performed with the topGO Bioconductor package (parameters: algorithm = “classic”, statistic = “fisher”). A gene was considered differentially expressed if the absolute log_2_ fold-change was greater than or equal to 3. Differentially expressed genes (DEGs) were functionally classified into KOG categories and subcategories on the basis of the manual curation of the results from the InterProScan analysis using InterProScan version 5.47–82.0 (Appendix A). Additionally, this table has been supplemented with the predicted homologs of each DEG identified in the following whole proteomes of *Schizosaccharomyces pombe* 972 h- [33], *Saccharomyces cerevisiae* S288C [34], *Candida albicans* SC5314 [35], *Cryptococcus neoformans* H99 [36], *M. circinelloides* f. *circinelloides* 1006PhL [37], *Neurospora crassa* OR74A [38], *Aspergillus fumigatus* Af293 [39], *Aspergillus nidulans* FGSC A4 [40], *Scedosporium apiospermum* IHEM 14462 [41], and *Anaeromyces robustus* S4 [42]. To this end, we grouped proteins of the above species into protein clusters by performing an all-vs-all search using MMseqs2 [43], setting sensitivity to 5.7 and coverage to 0.8. then, the Markov cluster algorithm [44,45] was used with an inflation parameter 2.0. Orthologs were found on the basis of reciprocal best hit search using MMSeqs2 with easy-rbh function.

## 3. Results

Our data revealed prominent changes in the gene expression when the culture has been transferred from aerobic to anaerobic conditions. The average number of raw sequence read pairs per sample was 3.95 million (3,603,894, 3,992,427, and 3,384,845, and 4,722,290, 3,349,089, and 4,617,836 reads in the aerobic and anaerobic sample triplicates, respectively) and 3.94 million (3,596,442, 3,984,122, and 3,376,382, and 4,715,087, 3,343,617, and 4,608,645 reads in the aerobic and anaerobic sample triplicates, respectively) after trimming. From the 11,791 annotated features in the applied transcriptome annotation, 9476 genes were expressed at the level of FPKM > 1 in all aerobic samples (9580, 9587, and 9578 expressed genes in the three replicates, respectively) and 9049 genes were expressed in all anaerobic samples (9259, 9249, and 9261 expressed genes in the three replicates, respectively); 8933 genes were expressed in all replicates of both conditions. In total, 539 DEGs (FDR < 0.05, |log_2_FC| ≥ 3) were identified, including 190 upregulated and 349 downregulated transcripts (Figure 1 and Appendix A). During functional characterization, DEGs were classified into four main functional categories. Among both down- and upregulated transcripts, the group of poorly characterized transcripts proved to be the most prominent, followed by transcripts related to “Metabolism”, “Cellular process and signaling”, and “Information storage and processing”.

Among metabolism-related genes, carbohydrate metabolism is affected the most during mycelium to yeast transition: 23 transcripts predicted to be involved in carbohydrate transport and metabolism were found to be upregulated, while 32 were downregulated. Among these transcripts, several encoded proteins were related to cell wall metabolism. For instance, transcripts encoding putative chitin deacetylases (ID 155630, 186155, 155048, 155566, and 156744) and predicted chitinases (108620 and 107467) were upregulated, while genes for other putative glycoside hydrolase/deacetylase enzymes (i.e., IDs 133575, 155735, 156524, 136560, 131995, and 155780), predicted chitin synthases (i.e., 151786 and 85917), and a putative chitinase (ID 105626) were found to be downregulated. Additionally, anaerobiosis also affected the transcription of transporters: among the 14 up- and 42 downregulated transporters, several putative sugar transporters were detected (Figure 2). Namely, we observed the upregulation of 155841, 151596, 145859, 105934, and 134811, which are putative sugar transporters belonging to the major facilitator superfamily (MFS) without any orthologs in the proteome of *S. pombe*, *S. cerevisiae*, *C. albicans*, or *A. nidulans*, and the downregulation of 153150 (an ortholog of a predicted transmembrane transporter in *A. nidulans* FGSC A4: AN2466 and a putative MFS glucose transporter in *C. albicans* SC5314 C1_02110C_ A), 155500 (an ortholog of *A. nidulans* FGSC A4: AN5104 with predicted transmembrane transporter activity), and 108626, among other provisionally classified proteins as carbohydrate transporters.

Additionally, a prominent number of transcripts related to amino acid transport and metabolism, lipid transport and metabolism, and energy production and conversion proved to be downregulated when the culture was transferred into an anaerobic atmosphere. In this latter group, transcripts related to oxidative metabolism were identified, such as predicted aconitases (i.e., 153298 and 186385) that catalyze citrate–isocitrate transition in TCA cycle, aldehyde dehydrogenases (i.e., 179978 and 157230), a predicted ATP-citrate lyase (i.e., 110808) that is potentially involved in the synthesis of Acetyl-CoA from citrate, predicted mitochondrial carrier proteins (e.g., 135605, 91563, 157132, 152871, 154970, and 155050), putative mitochondrial aldehyde dehydrogenases (e.g., 157230), putative glycerol-3-phosphate dehydrogenases (e.g., 153523, an ortholog of *A. nidulans* FGSCA4: GfdA), elements of the branched-chain α-ketoacid dehydrogenase complex (i.e., 139791 and 148011 encoding the E1 alpha and beta subunits, respectively, and 105619 encoding the E2 subunit), a putative mitochondrial external NADH dehydrogenase (i.e., 73711) that catalyzes the oxidation of cytosolic NADH, other oxidoreductases (e.g., 153534, 136103, and 122685), and a putative NADH-cytochrome b-5 reductase (i.e., 156238). Simultaneously, a small number of transcripts related to energy production and conversion were found to be upregulated, for instance, a predicted malate dehydrogenase (i.e., 186772), a putative acetyl-CoA hydrolase (i.e., 155896), a predicted fumarate reductase (104895), and a putative mitochondrial tricarboxylate/dicarboxylate carrier protein (i.e., 108484). In addition to this, we identified five predicted alcohol dehydrogenases (ADHs) among secondary metabolism-related DEGs: expression of 90838, 157439, and 160397 were downregulated, while that of 120424 and 155149 was upregulated under anaerobic conditions.

A relatively small number of transcripts with putative regulatory roles in dimorphism were also identified among the DEGs in our datasets. For example, a predicted PKAC subunit (47751) was found to be upregulated, while transcripts encoding a putative serine/threonine protein phosphatase 2A catalytic subunit (162179), a predicted cyclophilin-type peptidyl-prolyl cis-trans isomerase (154978), and a putative guanine nucleotide exchange factor (GEF; ProtID: 86922) were downregulated.

We also found 13 differentially upregulated and 18 downregulated transcription factors, and 12 up- and 20 downregulated carbohydrate-active enzymes (CAZymes) (Appendix A).

## 4. Discussion

Anaerobiosis, together with the presence by a fermentable hexose, induces the mycelium-to-yeast transition and a switch from respiration to fermentation in *Mucor* species [46]. Therefore, we expected that this condition would primarily affect the properties of the cell wall and the basic metabolic and energy conservation processes of the cell.

The fungal cell wall is a strictly regulated, dynamic structure, and its major physical and chemical properties, such as thickness, structure, and composition, are mainly regulated by the cell cycle and affected by various environmental conditions, such as temperature, pH, carbon source, and oxygen level [47]. According to the recent review of Lecointe et al. [48], the main components of the cell wall of Mucorales are chitin and its deacetylated form, chitosan, β-glucans, mucoran, and mucoric acid. The chitin content of the cell walls varies according to the morphological phase of the fungus: it was proven to be lower in the yeasts while higher in the filamentous fungi [49]. Additionally, compared to other fungal species, many Basidiomycota and Mucoromycota possess a much higher chitosan level relative to chitin [50]. For instance, in *C. neoformans*, the cell wall has minor quantities of chitin and higher quantities of chitosan, which, among others, helps to direct cell wall integrity and bud separation [51]. In contrast to this, the chitin and chitosan content of the cell wall of *Mucor rouxii* proved to be very similar in the yeast and the hyphal cultures [52]. The molecular mass of mucoran was reported to be almost two times higher in the mycelial form compared to the yeast-like cells [53,54]. Conversely, mucoric acid has a similar molecular mass in sporangiophores, mycelium, or yeast-like cells [55].

According to our DEG dataset, anaerobiosis did indeed affect the expression of several cell wall-related genes. Our analysis revealed the upregulation of transcripts encoding putative chitin deacetylases and predicted chitinases. This result could suggest an altered chitosan content in the cell wall of yeast cells under anaerobic growth, which may contribute to the production of multiple budding sites, as was previously demonstrated in *C. neoformans* [51]. According to López-Fernández et al. [56], the chitosan content affects the virulence properties of *M. lusitanicus*. Thus, the fact that yeast cells of *M. lusitanicus* are less virulent than its hyphal form [16], among others, might be explained by the different chitosan content of the cell wall.

Some of the various glycoside hydrolases/deacetylases, the predicted chitin synthases, and a putative chitinase were also downregulated in our dataset. It was previously demonstrated by Lopez-Matas et al. [57] that the transcript of *Mcchs1* (ID 114551) encoding a class II chitin synthase accumulates only during the exponentially growing hyphal stage, while it was not expressed in the yeast form. Although the transcript level of *Mcchs1* proved to be unchanged during the mycelium-yeast transition of *M. lusitanicus* in our study, differentially downregulated transcripts encoding other chitin synthases (i.e., 151786 and 85917) were detected. These results correspond with the assumption of Lopez-Matas et al. [57] that various chitin synthase activities may have different roles in the dimorphic growth of *Mucor* spp. Furthermore, the transcript level of a yet uncharacterized protein coding gene (ID 118900), which has a predicted role in “cell wall, membrane and envelope biogenesis” and shows sequence similarity with the *A. nidulans* FGSC A4: AN1554, a protein with a predicted role in the regulation of chitin synthase activity [58], was found to be decreased under anaerobic conditions. All these results mentioned above suggest that the dimorphic switch in *M. lusitanicus* is accompanied by changes in the chitin and chitosan content of the cell wall.

Anaerobic conditions induced marked changes in the expression of energy production- and conversion-related genes. As was expected, numerous transcripts related to oxidative metabolism (e.g., TCA cycle and oxidative phosphorylation) proved to be downregulated when the culture has been transferred into an anaerobic atmosphere. These findings agree with previous studies describing that *M. lusitanicus* switches from oxidative to fermentative metabolism under anaerobic conditions in the presence of hexoses, independently of its morphology [29]. In line with this, we identified five (i.e., three down- and two upregulated) predicted class V alcohol dehydrogenases (ADHs) among secondary metabolism-related DEGs, an enzyme that is responsible for the conversion of acetaldehyde into ethanol in *M. lusitanicus* [59]. These results suggest that alcoholic fermentation is not restricted to anaerobic cultures, and that ethanol production is differentially regulated in *M. lusitanicus* depending on the presence or absence of oxygen. Our observation corresponds with the previous description of *M. lusitanicus* as a Crabtree-positive microorganism. This means that it can produce ethanol under anaerobic and aerobic conditions as well. Aerobic ethanol production occurs when glucose concentration is sufficiently high in the environment, but once glucose is depleted, ethanol is consumed as a carbon source [25]. We identified a total of eight putative ADHs in the genome of *M. lusitanicus* CBS277.49 v2.0 (Protein IDs: 90838, 120424, 157439, 160397,155149, 139628, 140177, 152844, and 34200). Although their functions are unknown, four were differentially expressed in our dataset: 90838 and 160397 were found to be downregulated, while 120424 and 155149 were upregulated. Out of these transcripts, 155149 was previously characterized by Rangel-Porras et al. [59] as the major (if not the only) ADH encoded by *adh1* in *M. lusitanicus*, that was expressed in the cytosol of both mycelia and yeast cells. Adh1 has also been mentioned as a marker linked to fermentative metabolism under anaerobic growth in *M. lusitanicus* by Patiño-Medina et al. [60]. In *Mucor rouxii*, ADH activity was proven to be essential for anaerobic growth, most probably because of its necessity for the regeneration of NAD+ for glycolysis [61]. Furthermore, the higher transcript level of *adh1* in the yeast form of *M. lusitanicus* was also verified by the study of Valle-Maldonado et al. [14]. Moreover, as pyruvate is not just the key intermediate in sugar metabolism and a precursor for the synthesis of several amino acids, but also an intermediate for ethanol production, we assume that the upregulation of the predicted malate dehydrogenase 186772, an ortholog of *A. nidulans* FGSC A4: AN6168 (*MaeA*) with a predicted role in the conversion of malate to pyruvate, also correlates with the high levels of anaerobic ethanol production of *M. lusitanicus*. As a potential proof for the elevated ethanol production in the anaerobic culture, we also observed the upregulation of a putative acetyl-CoA hydrolase, which is found to be the ortholog of *A. nidulans* FGSC A4: AN1547 CoA-transferase (*CoaT*) with an ethanol- and acetate-inducible promoter [60].

Finally, other metabolism-related transcripts, such as 104895 and 187130 encoding a putative fumarate reductase and a multicopper oxidase Fet3A, respectively, were also upregulated in the absence of oxygen, suggesting that they might have a potential role in anaerobic growth. The fumarate reductase is the ortholog of the soluble fumarate reductase of *S. cerevisiae* S288C: YEL047C (FRD1), which has been proven to be required with isoenzyme Osm1p for anaerobic growth on glucose as a carbon source [61]. The *fet3a* has been recently characterized by Navarro-Mendoza et al. [62] as a member of a gene family of three ferroxidases. These authors observed that *Fet3A* was specifically expressed during yeast growth under anaerobic conditions, while *Fet3B* (ID 50174) and *Fet3C* (ID 91148) were expressed in the mycelium during aerobic growth [62]. In contrast to this, we did not observe the downregulation of either *Fet3B* or *Fet3C* in the anaerobic culture.

The role of cAMP-PKA signal transduction pathway in the dimorphism of *Mucor* species has been reported several times previously [26,28,47,63]. The recent work of Moriwaki-Takano et al. [64] provides the first detailed hypothetical scheme detailing the elements of this pathway and their possible interactions. Briefly, this work suggests that the intracellular bicarbonic ion (HCO^3−^) level is increased, suppressing the transcription of Nce103 (carbonic anhydrase), Ras3GTPase, and Pde (phosphodiesterase), while inducing the Cyr1 adenylate cyclase in *M. circinelloides* grown in CO_2_ atmosphere. These latter two components are assumed to be responsible for the high cAMP level in yeast cells. cAMP induces PKA activity, which is suggested to inhibit hyphal growth through the inhibition of the elongation factor Efg1 [64]. On the basis of this study, we expected the elements of the cAMP-PKA pathway to be differentially regulated under anaerobic conditions, but only a single PKAC subunit (47751)-encoding gene was proven to be upregulated under anaerobic conditions in our dataset. Interestingly, the expression of this subunit, PKAC3 has been investigated before and found to be upregulated after a 5 h shift from anaerobic yeast-like growth to aerobic hyphal growth [65]. It is worth mentioning that orthologues of *efg1* (122144) and *cyr1* (126447) were found to be down- and upregulated, respectively, with log_2_FC values ≤ 3, i.e., −0.998 and 0.572, respectively. Similarly, small changes in the expression levels of these genes were previously detected by Moriwaki-Takano et al. [64] after 6 h of cultivation under aerobic and anaerobic conditions.

Besides the cAMP-PKA pathway, calcineurin was proven to be a key regulator of dimorphism in *Mucor* species [15,16]. Investigating the expression of the calcineurin pathway components during the hyphal–yeast transition in *M. lusitanicus*, neither the catalytic and regulatory subunits of calcineurin, nor calmodulin or FKBP12-encoding transcripts were identified as differentially regulated.

In the present study, we investigated the transcriptional response of *M. lusitanicus* when transferred from aerobic to anaerobic atmosphere, and thus indirectly assessed the genes involved in the mycelium-to-yeast transition as well. While some of the transcriptional changes identified correspond well with previous published studies, our results also provide new valuable data to this field of research.

## Figures and Tables

**Figure 1 jof-08-00404-f001:**
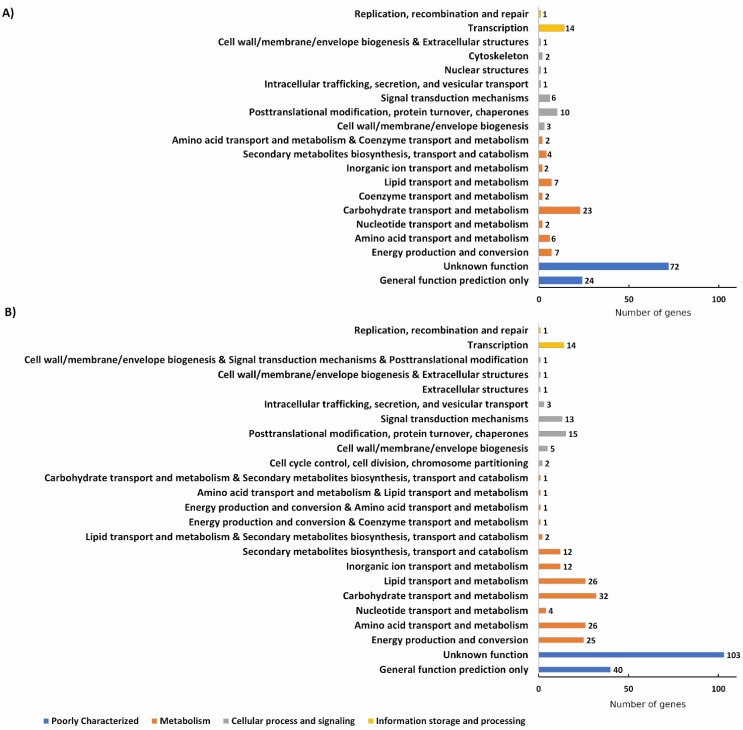
Differentially up- (**A**) and downregulated (**B**) genes of *M. lusitanicus* grown anaerobically compared to those after aerobic culturing of the fungus and their distribution among the main functional categories of the genes.

**Figure 2 jof-08-00404-f002:**
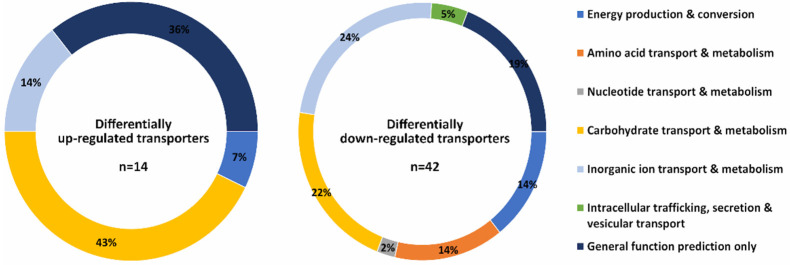
Proportion of the up- and downregulated transporter genes of *M. lusitanicus* grown anaerobically compared to those after aerobic culturing of the fungus.

## Data Availability

Trancriptomic datasets obtained during the study were deposited in the NCBI Sequence Read Archive (SRA) database (https://www.ncbi.nlm.nih.gov/sra) under the accession number PRJNA729254. Genes found to be differentially expressed (DEGs) are listed in Appendix A.

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
