# Peer review of "Differential Gene Expression of Mucor lusitanicus under Aerobic and Anaerobic Conditions"

_jof, 2022, doi:10.3390/jof8040404_

Round 1

Reviewer 1 Report

All comments and suggestions are in the attached document titled "Reviewer Comments"

Author Response

Answers to Reviewer 1

Thank you for the comments and suggestions. We are grateful for the thorough revision of the manuscript and pleased to accept all the suggested corrections. The changes made and the answers to the comments raised are as follows:

Line 89: It is stated that the samples are kept at -80°C, but it is not clear how these samples were collected.

Samples were filtered through a 0.45-um filter and RNA was extracted from the cells. This information has been added to the “2.2. RNA Extraction and Sequencing” section: “For RNA purification, the 30-ml cultures detailed above were filtered through 0.45 µm MCE membrane filters (Millipore) and the extraction was performed by using the Direct-zol™ RNA MiniPrep Kit (Zymo Research), following the recommendations of the manufacturer.”

Line 92: It is stated that 3 biological replicates per treatment condition were included, but were technical replicates used as well? If so, how many total samples were used to produce the dataset? For example, in the studies I am familiar with, 3 plates are pooled to make single replicate and three replicates are sequenced, thus 9 samples are used in total. This provides both biological and technical repeats and thus allows robust statistical analysis.

During the preparation of the samples, 2 technical and 3 biological, i.e., a total of 6 samples were sequenced. Biological samples were inoculated in duplicates in each case and assembled to obtain sufficient yeast-like cells. The corresponding sentence has been completed: “RNA-seq libraries were prepared from 3 biological and 2 technical replicates per treatment condition (i.e., 6 samples per treatment were used in total).”

Line 99: It is stated that libraries are pooled before sequencing- how many libraries were generated and how many were sequenced? 

Thank you for this comment. The following sentences were changed or added to the section: “RNA-seq libraries were prepared from three biological and two technical replicates per treatment condition (i.e., six samples per treatment were used in total). Thus, six libraries were generated and all of them were sequenced. Libraries were pooled before sequencing but since these were indexed libraries sequence reads belonging to different samples could be demultiplexed (separated to different files) after sequencing”.

Line 107: Is this genome publicly available through JGI? Many of their genomes have strict “no use until formally published” policies.

The genome is published. The missing citation has been added to the manuscript. (32) Corrochano, L.M.; Kuo, A.; Marcet-Houben, M.; Polaino, S.; Salamov, A.; Villalobos-Escobedo, J.M.; Grimwood, J.; Álvarez, M.I.; Avalos, J.; Bauer, D.; et al. Expansion of signal transduction pathways in fungi by extensive genome duplication. Curr. Biol. 2016, 26, 1577–1584, doi:10.1016/j.cub.2016.04.038.

Delete the first sentence of the results- it’s a summary of the methods section and is unnecessary.

This sentence has been deleted.

It would be helpful to divide your results into discrete sub-sections.

Thank you for this suggestion. We think that, after reformulating many parts according the suggestions of the five reviewers, the clarity of this section has improved and we can maintain the compactness of the text and its form of Communication.

Line 102: This sentence belong in the results section, with other related data. Please include a section in the results that details the RNA seq statistics, including number of reads generated per library, how many reads were present after trimming and QC, how many genes were shown to be expressed under each condition etc. 

The sentence has been transferred from the Materials and methods to the Results and completed with the requested data as follows: “The average number of raw sequence read pairs per sample was 3.95 million (3603894, 3992427 and 3384845, and 4722290, 3349089 and 4617836 reads in the aerobic and anaerobic sample triplicates, respectively) and 3.94 million (3596442, 3984122 and 3376382, and 4715087, 3343617 and 4608645 reads in the aerobic and anaerobic sample triplicates, respectively) after trimming. From the 11791 annotated features in the applied transcriptome annotation, 9476 genes were expressed at the level of FPKM>1 in all aerobic samples (9580, 9587 and 9578 expressed genes in the three replicates, respectively) and 9049 genes were expressed in all anaerobic samples (9259, 9249 and 9261 expressed genes in the three replicates, respectively); 8933 genes were expressed in all replicates of both conditions.”

Table 1 seems unnecessary given that it takes up a significant amount of space while CAZymes are not a major part of the results. I suggest either moving the table to the supplementary data, writing a more detailed results section on the various CAZymes that were differentially expressed, or creating a table that highlights the various DEGs (including CAZymes and other proteins)

Table 1 has been deleted as the data of the CAZymes are also in Table S1.

Line 233 - 236: The relevance of the two references here is not clear, perhaps you can reword this sentence to emphasize their importance.

The sentences have been reformulated as follows: “According to López-Fernández et al. [56], the chitosan content affects the virulence properties of M. lusitanicus. Thus, the fact that yeast cells of M. lusitanicus are less virulent than its hyphal form [16], among others, might be explained by the different chitosan content of the cell wall.”

Line 241 - 243: This sentence on Mcchs1 is confusing and I’m not really sure what it means. For example: how can the other chitin synthases be homologous to Mcchs1? Has there been a gene duplication or do you just mean that their functions many be similar and thus although Mcchs1 is not differentially expressed, the chitin synthesis pathway has been regulated? Please reword this sentence.

The sentence has been reworded as follows: “It was previously demonstrated by Lopez-Matas et al. [57] that the transcript of Mcchs1 (ID 114551) encoding a class II chitin synthase accumulates only during the exponentially growing hyphal stage, while it was not expressed in the yeast form. Although the transcript level of Mcchs1 proved to be unchanged during the mycelium-yeast transition of M. lusitanicus in our study, differentially downregulated transcripts encoding other chitin synthases (i.e., 151786 and 85917) were detected.”

Line 262 - 266: This is a really long sentence and is quite confusing to follow. Please can you shorten/split it and reword it to be clearer.

This sentence has been reformulated as follows: “Our observation corresponds with the previous description of M. lusitanicus as a Crabtree-positive microorganism. This means that it can produce ethanol under anaerobic and aerobic conditions, as well. Aerobic ethanol production happens when glucose concentration is sufficiently high in the environment, but once glucose is depleted, ethanol is consumed as a carbon source [25].”

Line 267 - 279: This section is difficult to follow and would benefit from some rewriting. Perhaps it would be better if you started the section with “We identified a total of eight putative ADHs in the genome of M. lusitanicus and although their functions are unknown, two were differentially expressed in our dataset”.

This part has been reformulated as follows: “We identified a total of eight putative ADHs in the genome of M. lusitanicus CBS277.49 v2.0 (Protein IDs: 90838, 120424, 157439, 160397,155149, 139628, 140177, 152844 and 34200). Although their functions are unknown, four were differentially expressed in our dataset: 90838 and 160397 were found to be downregulated, while 120424 and 155149 were upregulated. Out of these transcripts, 155149 was previously characterized by Rangel-Porras et al. [57] as the major (if not the only) ADH encoded by adh1 in M. lusitanicus, that was expressed in the cytosol of both mycelia and yeast cells. Adh1 has also been mentioned as a marker linked to fermentative metabolism under anaerobic growth in M. lusitanicus by Patiño-Medina et al. [58]. In Mucor rouxii, ADH activity proved to be essential for anaerobic growth, most probably, because of its necessity for the regeneration of NAD+ for glycolysis [59]. Furthermore, the higher transcript level of adh1 in the yeast form of M. lusitanicus was also verified by the study of Valle-Maldonado et al. [13].”

Line 300 - 341: Please provide possible reasons why the numerous genes involved in the PKA calcineurin pathways weren’t shown to be differentially regulated in your dataset. Given the importance of both these pathways in the morphological switch, this result is surprising and deserves a thorough explanation as to why they were not detected in a transcriptomic study.

Concerning this question, the following information has been added to this paragraph: “It is worth to mention that orthologues of efg1 (122144) and cyr1 (126447) found to be down and upregulated, respectively, with log2FC values ≤3, i.e., -0.998 and 0.572, respectively. Similarly small changes in the expression levels of these genes were previously detected by Moriwaki-Takano et al. [64] after 6 h of cultivation under aerobic and anaerobic conditions.”

Please also provide a concluding remarks section as the current discussion ends rather abruptly and unexpectantly. Perhaps include potential future research or how the information gained from this study could be used in the future.

Thank you for this suggestion. The following paragraph has been added to the end of the Discussion: “In the present study, we investigated the transcriptional response of M. lusitanicus when transferred from aerobic to anaerobic atmosphere, and thus, indirectly assessed the genes involved in the mycelium to yeast transition as well. While some of the transcriptional changes identified correspond well with previous published studies, our results also provide new valuable data to this field of research.”

Grammatical/typographical Issues: 

All suggested and listed changes have been made in the revised version.

We are grateful for the Reviewer's valuable comments and suggestions. A new version of the manuscript has been uploaded. We used "tracked changes mode" to highlight all corrections made in the Manuscript file. We hope that the revised manuscript will be acceptable for publication.

Yours sincerely,

Tamas Papp

Reviewer 2 Report

General comments:

The manuscript "Differential gene expression of Mucor lusitanicus under aerobic and anaerobic conditions" by Homa et al. brings new data and is of broad interest for the scientific community. Indeed, it indirectly yields data regarding gene regulation occurring during the mycelium to yeast transition, which could be (i) an important phenomenon regarding our understanding of the virulence in dimorphic Mucorales and (ii) may serve as a morphogenetic model. 

I have no major concerns regarding the sound results obtained and some minor comments are reported below:

1- Line 35: It could be of interest to cite the study which allow to consider the different members of the M. circinelloides complex as discrete species (Wagner et al., 2020).

2- Line 133 and line 146: It is noteworthy that dentification of genes involved in the mycelium to yeast transition was indirectly assessed since the overall effect is due to the transition toward anaerobiosis.

3- Line 220, I would avoid Zygomycetes and use instead Mucoromycota

4- Text in Figure 1 is hardly readable and although discussion of the different DEGs is well structured in the text and not too descriptive, the authors might consider a differentially expressed genes network analysis and I would have appreciated a diagram focusing on the impact of the gene regulation observed at least on cell wall composition and on energy production. 

Author Response

Response to Reviewer 2

Thank you for your comments, we are grateful for the thorough revision of our manuscript. We have addressed all the comments as explained below.

1- Line 35: It could be of interest to cite the study which allow to consider the different members of the M. circinelloides complex as discrete species (Wagner et al., 2020).

The citation has been added to the manuscript, as follows: “Mucor lusitanicus (also known as M. circinelloides f. lusitanicus) is a filamentous fungus belonging to the order Mucorales [1].”

Wagner, L.; Stielow, J.B.; de Hoog, G.S.; Bensch, K.; Schwartze, V.U.; Voigt, K.; Alastruey-Izquierdo, A.; Kurzai, O.; Walther, G. A new species concept for the clinically relevant Mucor circinelloides complex. Persoonia Mol. Phylogeny Evol. Fungi 2020, 44, 67–97, doi:10.3767/persoonia.2020.44.03.

2- Line 133 and line 146: It is noteworthy that dentification of genes involved in the mycelium to yeast transition was indirectly assessed since the overall effect is due to the transition toward anaerobiosis.

This sentence has been deleted as requested by Reviewer 1.

3- Line 220, I would avoid Zygomycetes and use instead Mucoromycota

Thank you for this comment. The term Zygomycetes has been changed to Mucoromycota as follows: „Additionally, compared to other fungal species, many Basidiomycota and Mucoromycota possess a much higher chitosan level relative to chitin”

4- Text in Figure 1 is hardly readable and although discussion of the different DEGs is well structured in the text and not too descriptive, the authors might consider a differentially expressed genes network analysis and I would have appreciated a diagram focusing on the impact of the gene regulation observed at least on cell wall composition and on energy production.

We reedited this figure for better reading and understanding, according the request of the Reviewers.

We are grateful for the Reviewer's valuable comments. A new version of the manuscript has been uploaded. We used "tracked changes mode" to highlight all corrections made in the Manuscript file. We hope that the revised manuscript will be acceptable for publication.

Yours sincerely,

Tamas Papp

Reviewer 3 Report

This work tried to describe the influence of the dimorphism on the global transcriptional framework. Although interesting, there are some issues of concern

But  my main concerns are:

First and the most important  the concept of dimorphism in Mucor lusitanicus in order to interpret the results

The authors experimentally established the global transcript levels in mycelial growth from 24 h obtained from sporangiospores and compared those transcript levels respect to the transcripts form yeast that were obtained from hyphae-yeast transition from 4 h of growth.

First. the author must establish and compare  the transcript levels from hyphae and yeast obtained both from sporangiospores (please refer as spores). Because in that case both vegetative development (yeast or mycelial) are derived from the same starting cells (spores), and the transcriptional regulation could be different between yeast from sporangiospores and yeast that are obtained from hyphae-yeast transition. Otherwise, the transcriptional changes that the authors described could be misinterpreted, for example: Are the changes in the transcript levels between mycelia and yeast (obtained from hyphae-yeast transition)  that the authors described  involved in the transition? or in yeast development? or in both processes?

Otherwise it is very confusing the interpretation of the results.

Second. why did the authors decide  to analyze and compare 24 h of mycelial growth respect to 4 h for yeast (obtained from hyphae-yeast transition)?

To be comparable both vegetative growth, the authors must compare similar phases of growth, for example exponential hyphae growth versus exponential yeast growth, otherwise again, how could the results be interpreted? the differences correspond to differences in the phases of growth.

For 24 hours of growth, the mycelial is in the stationary phase and is probably that some of the mycelia could be in the decay phase. It is already described that mycelial has a tendency to develop arthorspores under nutritional deficiency. Also it is probably that some nutrients as the glucose could be depleted after 24 h of mycelial growth.

Meanwhile,  4 hours of growth after the hyphae-yeast transition, the yeast start to appear at this time and for sure the nutrient condition will be different compared to 24 h of mycelial growth. So the transcriptional changes that the authors claim are due the nutrient/phase of growth or yeast development?

Third.  YNB (yeast nitrogen base) media supplemented with 1% of glucose was used for mycelial growth and mycelial-yeast transition.  Since Orlowski review (1991), the yeast development of some species of Mucor as racemosus (and also in lusitanicus) need an organic nitrogen source for an adequate yeast development. In other words, although M. lusitanicus (as racemosus and circinelloides) can growth under anaerobic condition with ammonium sulphate (YNB) as only nitrogen source, this nitrogen source  is not the most  adequate for yeast development, in fact  a mixture of hyphae and yeast are obtained. For this reason YPG is better in this aspect. which contain peptone as nitrogen source.

Why to use 1% of glucose when normally several groups used 2%, is there a reason?

So, in general this issues are confusing for the interpretation of the results.

i considered 

Author Response

Response to Reviewer 3

We are grateful for the Reviewer's valuable comments. Our answers to them raised are as follows:

First. the author must establish and compare  the transcript levels from hyphae and yeast obtained both from sporangiospores (please refer as spores). Because in that case both vegetative development (yeast or mycelial) are derived from the same starting cells (spores), and the transcriptional regulation could be different between yeast from sporangiospores and yeast that are obtained from hyphae-yeast transition. Otherwise, the transcriptional changes that the authors described could be misinterpreted, for example: Are the changes in the transcript levels between mycelia and yeast (obtained from hyphae-yeast transition)  that the authors described  involved in the transition? or in yeast development? or in both processes?

Thank you for this comment. Our intention was to examine the transcriptional changes when the environment of a colony changes from aerobic to anaerobic. The colony was grown under aerobiosis and was transferred into an anaerobic environment. We examined the effect of the transition of the fungus into the anaerobic environment. So, in first line, the transcriptomic data reflects to the switch of the growth and metabolism from aerobiosis to anaerobiosis. As the fungus grow in a yeast form under anaerobiosis and yeast-like cells are present after four h in oxygen free environment, it can be assumed that the transcriptome also reflects to the switch from the mycelial to the yeast growth (in a second line). This short communication would give a first picture about the metabolic and morphological changes, which follow the environmental changes, at a transcriptional level.

If we would grow the fungus from the spores (i.e., in the whole life cycle) under either aerobiosis and anaerobiosis, we would detect the genes, which are necessary during the growth under these conditions and not those, which are involved into the transition. Certainly, it is also an interesting and important issue, which can be worth investigating in the future.

Second. why did the authors decide  to analyze and compare 24 h of mycelial growth respect to 4 h for yeast (obtained from hyphae-yeast transition)?

We wanted to study genes that are activated/inhibited during the transition. Previously, we observed when the transition began and chose the time to focus on the processes at the beginning of the transition. The 24 h preculture was to have a cell volume that could be analyzed (e.g., by RNA extraction). Four h was chosen because yeast cells were already present at that time point.

To be comparable both vegetative growth, the authors must compare similar phases of growth, for example exponential hyphae growth versus exponential yeast growth, otherwise again, how could the results be interpreted? the differences correspond to differences in the phases of growth.

As described above, it would be interesting to compare colonies, which are in the same phases. However, in this case, we cannot examine, which genes are activated or repressed under the change of the metabolism.

For 24 hours of growth, the mycelial is in the stationary phase and is probably that some of the mycelia could be in the decay phase. It is already described that mycelial has a tendency to develop arthorspores under nutritional deficiency. Also it is probably that some nutrients as the glucose could be depleted after 24 h of mycelial growth.

The low cell count and large amount of nutrient solution, in our opinion, precludes the possibility of glucose starvation. In addition, the yeast transition requires a sufficient concentration of glucose, besides the lack of oxygen. Thus, the fact that yeast cells were observed contradict with the glucose starvation. Arthrospore formation was not considered as yeast formation or transition in our experiment. Only single cell, spherical, budding cells were considered as the yeast-like cells - only their presence was considered as the transition and they were used for RNA extraction.

Meanwhile,  4 hours of growth after the hyphae-yeast transition, the yeast start to appear at this time and for sure the nutrient condition will be different compared to 24 h of mycelial growth. So the transcriptional changes that the authors claim are due the nutrient/phase of growth or yeast development?

The nutrient condition could be somewhat different but most transcriptional changes can be responses primarily to the lack of oxygen and secondly to the morphological changes and less to the nutrient level. This is reinforced by the types of the differentially expressed genes.

Third.  YNB (yeast nitrogen base) media supplemented with 1% of glucose was used for mycelial growth and mycelial-yeast transition.  Since Orlowski review (1991), the yeast development of some species of Mucor as racemosus (and also in lusitanicus) need an organic nitrogen source for an adequate yeast development. In other words, although M. lusitanicus (as racemosus and circinelloides) can growth under anaerobic condition with ammonium sulphate (YNB) as only nitrogen source, this nitrogen source  is not the most  adequate for yeast development, in fact  a mixture of hyphae and yeast are obtained. For this reason YPG is better in this aspect. which contain peptone as nitrogen source.

Thank you this comment. We have used YNB several times to obtain yeast cells under anaerobiosis and it has always “worked”. Using this medium we can grow cultures containing purely yeast cells.

Why to use 1% of glucose when normally several groups used 2%, is there a reason?

We routinely use this concentration as it has been documented in several publications (for example, Fungal Genet Biol 2019. 129, 30; Int J Mol Sci 2020. 21, 3727; Front Cell Infect Microbiol 2021. 11, 273). However, we tested the 2% of glucose and we did not observe differences in the hyphal-yeast transition and growth.

So, in general this issues are confusing for the interpretation of the results.

We reedited the text and reformulated several parts (including the figures and tables) according the suggestions of the five Referees reviewed the manuscript. We hope that clarity and informativeness of our manuscript has improved.

We are grateful for the Reviewer's valuable comments. A new version of the manuscript has been uploaded. We used "tracked changes mode" to highlight all corrections made in the Manuscript file. We hope that the revised manuscript could be acceptable for publication.

Yours sincerely,

Tamas Papp

Reviewer 4 Report

The following are my comments and critique should be addressed before acceptance for publication:

  • All words in “Keywords” should not be in bold
  • Line 36, species not genus
  • “Certain species, such as Rhizopus and Mucor , “ Its wrong, the paper contains some problems related to taxonomy “species and/or genus”. Please, check and correct the systematic and classification of mentioned organisms as “systematicroles”
  • Please, provide more detailed on culture conditions, 7 days ?, pH ? volume ? shaking “rpm” ? etc..
  • Was the RNA extractionand related experiments were conducted by a method from the author laboratory or taken from literature? There are no references mentioned for these methods.
  • Line, 129. make a space before setting.
  • Figure 1. I think its need to improve the quality
  • Define abbreviations upon first appearance in the textspecially name of microbes
  • I recommend the author to check the reference list for format. Some references are lacking information andstyle.

Author Response

Response to Reviewer 4

Thank you for the comments and suggestion. We are grateful for the revision and pleased to accept the suggested corrections. The changes made and the answers to the comments raised are as follows:

All words in “Keywords” should not be in bold

Corrected as requested.

Line 36, species not genus

The sentence has been changed as follows: “Several Mucor species have biotechnological significance as producers…”

 “Certain species, such as Rhizopus and Mucor , “ Its wrong, the paper contains some problems related to taxonomy “species and/or genus”. Please, check and correct the systematic and classification of mentioned organisms as “systematicroles”

The sentence has been changed as follows: “Certain species, such as members of the genera Rhizopus and Mucor, are also known as…”

Please, provide more detailed on culture conditions, 7 days ?, pH ? volume ? shaking “rpm” ? etc..

Data have been provided as requested and the section “2.1. Strain and Culture Conditions” has been completed as follows: “The M. lusitanicus double auxotrophic strain MS12 (leuA- and pyrG-), which is a derivative of the strain CBS 277.49 [31] was used in the present study. For spore collection, the strain was grown for 7 days on malt extract agar (MEA; 1% glucose, 0.25% yeast extract, 5% malt extract, 2% agar, pH 5.4) plates at 25 °C, then, the spores were washed with PBS by gentle scraping, and the spore suspensions obtained were maintained at 4 °C until use. For aerobic growth, 30 ml liquid minimal medium (YNB; 1% glucose, 0.15% ammonium sulphate, 0.15% sodium glutamate, 0.05% yeast nitrogen base without amino acids (Difco), supplemented with 0.05% leucine and 0.05% uracil, pH 6.8) was inoculated to get a final concentration of 104 sporangiospores/ml. The cultures were kept at 25 °C for 24 h with constant shaking (i.e., 190 rpm).”

Was the RNA extractionand related experiments were conducted by a method from the author laboratory or taken from literature? There are no references mentioned for these methods.

We used the given kit for RNA extraction according the manufacturer’s recommendations. The method description ahs been completed: “For RNA purification, the 30-ml cultures detailed above were filtered through 0.45 µm MCE membrane filters (Millipore) and the extraction was performed by using the Direct-zol™ RNA MiniPrep Kit (Zymo Research), following the recommendations of the manufacturer.”

Line, 129. make a space before setting.

Corrected.

Figure 1. I think its need to improve the quality

Figure 1 has been completely reedited for better quality and readability.

Define abbreviations upon first appearance in the text specially name of microbes

The text has been checked and corrected.

I recommend the author to check the reference list for format. Some references are lacking information and style.

The reference list has been corrected.

We are grateful for the Reviewer's valuable efforts to improve our manuscript. A new version of the manuscript has been uploaded. We used "tracked changes mode" to highlight all corrections made in the Manuscript file. We hope that the revised manuscript can be acceptable for publication.

Yours sincerely,

Tamas Papp

Reviewer 5 Report

This study by Homa et all addresses the differential gene expression of Mucor lusitanicus under aerobic and anaerobic growth conditions. It presents interesting results and I have only some minor comments, that are given below.

Overall, the manuscript is well-written, but it would benefit from a slight edit for grammar and style.

- do not abbreviate the genus name at the beginning of sentences (e.g. line 15, line 48)

- line 15: delete filamentous

- line 72: differentially activated? Isn’t it preferable to use expressed instead of activated?

- It is not clear, not for me at least, why a double auxotroph strain was used instead of a wild type strain?

- line 107: CBS277.49 is a strain of M. lusitanicus, correct? It should be clearly stated here. Also, maybe it is better to cite the original publication related to this genome (Curr Biol. 2016 Jun 20;26(12):1577-1584. doi: 10.1016/j.cub.2016.04.038) as requested by JGI.

- line 116: Ontology

- Figure 1 as it is does not work well. The text in the image is impossible to read.

- line 241-243: this section is not entirely clear. Which other chitin synthase Mcchs1 homologs? From class II? Or are you referring to other chitin synthase classes?

Also, the difference to previous results by Lopez-Matas et al. should be further discussed.

- line 248-249: In fact, this does not verify anything. At best it suggests but needs to be further confirmed by analysing the cell wall composition.

- line 263: Crabtree

- line 319-323: This part seems somehow incomplete. This transcript identified is relevant why?

- Finally, I miss a final concluding paragraph to wrap up everything. The discussion ends rather abruptly.

Author Response

Response to Reviewer 5

We are grateful for the Reviewer's valuable suggestions. We used "tracked changes mode" to highlight all corrections made in the Manuscript file. We hope that the revised version can now be accepted for publication. The changes made and the answers to the comments raised are as follows:

- do not abbreviate the genus name at the beginning of sentences (e.g. line 15, line 48)

Corrected.

- line 15: delete filamentous

Deleted.

- line 72: differentially activated? Isn’t it preferable to use expressed instead of activated?

The term “activated” has been changed to “expressed”.

- It is not clear, not for me at least, why a double auxotroph strain was used instead of a wild type strain?

We added the information to the manuscript that MS12 is a derivative of the sequenced, wild-type CBS 277.49 (see first sentence of the Materials and methods: “The M. lusitanicus double auxotrophic strain MS12 (leuA- and pyrG-), a derivative of the strain CBS 277.49 [31] was used in the present study.”). It was used because it is a model organism routinely used for genetic manipulation of Mucor (for example, Fungal Genet Biol 2019. 129, 30; Int J Mol Sci 2020. 21, 3727; Front Cell Infect Microbiol 2021. 11, 273). We understand that a prototrophic strain can be more appropriate to examine the anaerobiosis or morphological switch, but we thought that this communication about the genetic and expression patterns in this model strain can be interesting and informative for those who use this or other strains of Mucoralean fungi.

- line 107: CBS277.49 is a strain of M. lusitanicus, correct? It should be clearly stated here. Also, maybe it is better to cite the original publication related to this genome (Curr Biol. 2016 Jun 20;26(12):1577-1584. doi: 10.1016/j.cub.2016.04.038) as requested by JGI.

It is correct. At the time of the publication of the CBS 277.49 genome, it was classified as M. circinelloides f. lusitanicus but recently this subspecific taxon has been described as a species, i.e., M. lusitanicus (Wagner et al. 2020 Persoonia 44, 67 doi:10.3767/persoonia.2020.44.03.). This reference was also added to the manuscript (ref. 1 in the list). The cited website of the genome also names CBS 277.49 as M. lusitanicus. The requested reference was added to the text, and M. lusitanicus is mentioned as follows: “Sequence reads were aligned to the M. lusitanicus CBS 277.49 v2.0 reference genome assembly available from the Joint Genome Institute (JGI) website (https://genome.jgi.doe.gov/Mucci2/Mucci2.home.html [32]”

- line 116: Ontology

Corrected.

- Figure 1 as it is does not work well. The text in the image is impossible to read.

Figure 1 has been reedited for better readability.

- line 241-243: this section is not entirely clear. Which other chitin synthase Mcchs1 homologs? From class II? Or are you referring to other chitin synthase classes?

Also, the difference to previous results by Lopez-Matas et al. should be further discussed.

This section has been revised based on the suggestions of Reviewers 1 and 5 as follows: “Some of the various glycoside hydrolases/deacetylases, the predicted chitin synthases, and a putative chitinase were also downregulated in our dataset. It was previously demonstrated by Lopez-Matas et al. [57] that the transcript of Mcchs1 (ID 114551) encoding a class II chitin synthase accumulates only during the exponentially growing hyphal stage, while it was not expressed in the yeast form. Although the transcript level of Mcchs1 proved to be unchanged during the mycelium-yeast transition of M. lusitanicus in our study, differentially downregulated transcripts encoding other chitin synthases (i.e., 151786 and 85917) were detected. These results correspond with the assumption of Lopez-Matas et al. [57], that various chitin synthase activities may have different roles in the dimorphic growth of Mucor spp.”

- line 248-249: In fact, this does not verify anything. At best it suggests but needs to be further confirmed by analysing the cell wall composition.

This statement has been refined as follows: “All these results mentioned above suggest that the dimorphic switch in M. lusitanicus is accompanied by changes in the chitin and chitosan content of the cell wall.”

- line 263: Crabtree

Corrected

- line 319-323: This part seems somehow incomplete. This transcript identified is relevant why?

Thank you for this comment. These sentences were removed, since they remained mistakenly in the submitted version of the manuscript.

- Finally, I miss a final concluding paragraph to wrap up everything. The discussion ends rather abruptly.

A final concluding paragraph has been added to the Discussion as follows: “In the present study, we investigated the transcriptional response of M. lusitanicus when transferred from aerobic to anaerobic atmosphere, and thus, indirectly assessed the genes involved in the mycelium to yeast transition as well. While some of the transcriptional changes identified correspond well with previous published studies, our results also provide new valuable data to this field of research.”

We are grateful for the Reviewer’s efforts to improve the manuscript. A revised version of the manuscript was submitted. We used "tracked changes mode" to highlight all corrections made in the Manuscript file. We hope that the revised version can now be accepted for publication.

Yours sincerely,

Tamas Papp

Round 2

Reviewer 3 Report

I am agree with the changes made by the autors, please revise your references it seems that some are missing

Author Response

Response to Reviewer 3

Thank you for the comments.

We checked and corrected the references in the text and in the reference list too, as suggested.

We are grateful for the Reviewer's valuable comments and suggestions. A new version of the manuscript has been uploaded. We used "tracked changes mode" to highlight all corrections made in the Manuscript file. We hope that the revised manuscript will be acceptable for publication.

Yours sincerely,

Tamas Papp